# Effects of Compound Microecological Preparation Supplementation on Production Performance and Nutrient Apparent Digestibility in Hu Sheep from the Rumen Perspective

**DOI:** 10.3390/microorganisms13050999

**Published:** 2025-04-27

**Authors:** Mu-Long Lu, Long Pan, Chen Zheng, Ruo-Yu Mao, Guo-Hong Yuan, Chen-Yang Shi, Zhe-Huan Pu, Hui-Xin Su, Qi-Yu Diao, Halidai Rehemujiang, Gui-Shan Xu

**Affiliations:** 1College of Animal Science and Technology, Tarim University, Alar 843300, China; a2774148614@163.com (M.-L.L.); 13309359388@163.com (G.-H.Y.); 15809000246@163.com (C.-Y.S.); 13704202703@163.com (Z.-H.P.); huixinsu@163.com (H.-X.S.); 2College of Animal Science and Technology, Nanjing Agricultural University, Nanjing 210095, China; panlong@njau.edu.cn; 3College of Animal Science and Technology, Gansu Agricultural University, Lanzhou 730070, China; zhengc@gsau.edu.cn; 4Gene Engineering Laboratory of Feed Research Institute, Key Laboratory of Feed Biotechnology of the Ministry of Agriculture and Rural Affairs, Chinese Academy of Agricultural Sciences, Beijing 100080, China; maoruoyu@caas.cn; 5Key Laboratory of Livestock and Forage Resources Utilization Around Tarim, Ministry of Agriculture and Rural Affairs, Tarim University, Alar 843300, China; 6Institute of Feed Research, Key Laboratory of Feed Biotechnology of the Ministry of Agriculture and Rural Affairs, Chinese Academy of Agricultural Sciences, Beijing 100080, China; diaoqiyu@caas.cn

**Keywords:** antimicrobial peptides, *Saccharomyces boulardii*, enzyme activity, rumen microorganisms, *Ruminococcus*

## Abstract

This study evaluates the effects of a compound microecological preparation named ATABG, which is composed of antimicrobial peptide ID13 and *Saccharomyces boulardii*, on Hu sheep’s growth performance, feed digestibility, and rumen parameters. A total of 40 three-month-old Hu sheep (21.65 ± 0.33 kg) were randomly assigned to two groups: the control group (Con), which received a basal diet, and the experimental group (ATABG), which received the same diet supplemented with 1 g/kg ATABG on a dry matter basis. After a 10-day pre-feeding period to adapt the animals to the experimental diet, dry matter intake and weight gain were recorded during the subsequent 63-day trial period. Body weight was measured on days 1, 21, 42, and 63 of the trial, and animals were slaughtered on day 63 to collect rumen fluid and tissue. Results indicated that ATABG supplementation significantly increased the apparent digestibility of crude protein, neutral detergent fiber, acid detergent fiber, and organic matter (*p* < 0.05). Rumen fluid analysis revealed increased microbial protein concentration and cellulase activity (*p* < 0.05) in the ATABG group. Microbiota analysis indicated that ATABG increased the relative abundance of *Ruminococcus* and Proteobacteria, elevated Firmicutes, and reduced Bacteroidota (*p* < 0.05). Correlation analysis showed *Ruminococcus* was positively associated with crude protein digestibility, while *Quinella* correlated with growth-related indices (r > 0.4, *p* < 0.05). In conclusion, ATABG supplementation improves protein digestibility and rumen microbial protein synthesis by enriching *Ruminococcus* and enhancing cellulase activity, potentially optimizing nitrogen utilization in Hu sheep.

## 1. Introduction

Ruminants are a significant source of animal protein in the human diet, particularly through milk and meat [1]. Their digestive process is intricate and relies on the synergistic action of various microorganisms and enzymes in the rumen; this synergy enables the extraction of more energy and nutrients from plant materials [2]. Factors such as breed, feed, age, gender, and stress influence the ruminants’ digestion and productivity [3,4]. Concurrently, although antibiotics effectively enhance livestock performance and immunity, their use has been increasingly phased out of the production process due to concerns about residues. Consequently, a shift has been made toward exploring more natural, residue-free methods to enhance livestock performance [1].

Antimicrobial peptides (AMPs) are currently being evaluated as potential alternatives to antibiotics [5]. Existing research on AMPs focuses on the effects on immunity and antioxidant indexes, but less on the effects of antimicrobial peptides on the production and digestion induced by changes in the structure of the microbial community in the digestive tract [6]. The study by Shi et al. [7] proved that antimicrobial peptides could improve the bull’s production performance, digestive enzymes, and fermentation parameters. Previous studies have assessed the designed peptide named ID13 in terms of its bioavailability, structure, as well as its anti-*Staphylococcus aureus* mechanism and pharmacodynamics [5,8]. However, there is scarce information about the application of ID13 in non-model animals, and specific experiments are necessary to evaluate its effect in actual production.

Numerous strategies exist for enhancing feed digestibility, including physical processing [9,10], chemical treatment [11], and microbiological treatment [12,13]. Probiotics, also referred to as direct-fed microbials, are live microbial strains that can provide health and nutritional benefits to the host when administered in appropriate amounts [1]. Among them, bacterial, fungal, and yeast probiotics have been extensively utilized in ruminants, demonstrating effectiveness in enhancing various physiological functions [1]. Previous research on probiotics, specifically *Saccharomyces boulardii* (*S. boulardii*) has shown that this yeast can improve feed intake and milk yield in lactating dairy cows [14]. An increase in the overall population of *Lactobacilli* was noted in the rumen of weaned calves supplemented with *S. boulardii*, as reported by Fomenky et al. [15]. In a study conducted by Roos et al. [16], the addition of *S. boulardii* showed promising vaccine efficacy, evidenced by significantly increased seroconversions against bovine herpesvirus 5 (*p* > 0.001). *S. boulardii* is characterized by several key traits, including resistance to high temperatures, acidity, bile salts, and antioxidant activity [17]. These traits enable it to survive and function effectively within the challenging environment of the livestock digestive tract. However, few studies have focused on the impact of *S. boulardii* on growth performance and apparent digestibility from the rumen perspective in sheep.

In this experiment, we examined the compound microecological preparation named ATABG, which is composed of AMPs named ID13 [8] and *S. boulardii*. Our innovation, compared to previous studies, lies in the unique combination of these components within ATABG. Furthermore, our focus extends beyond performance and digestibility; we also conducted a comprehensive analysis of the rumen environment, assessing parameters such as rumen pH, ammonia nitrogen (NH_3_-N), volatile fatty acid (VFA), digestive enzyme activities, and rumen bacterial populations. Based on prior research findings related to similar components or animal models, we hypothesize that the inclusion of ATABG will influence the composition of rumen microbiota in Hu sheep. This alteration may enhance feed digestibility and feed efficiency, ultimately affecting the production performance of Hu sheep. To test this hypothesis, we investigated changes in growth performance, nutritional efficiency, and rumen parameters, along with their interactions with rumen bacterial abundance in Hu sheep.

## 2. Materials and Methods

### 2.1. Animal Ethics

Vaccinated sheep were purchased from a Hu sheep farm in Alar, Xinjiang (40.54° N, 81.29° E). All the experimental procedures applied in this study were reviewed and approved by the Animal Ethics Committee of Tarim University (approval number 2024068).

### 2.2. Blended Feed Additives

ATABG is a complex composed of bioactive peptides and modified *S. boulardii* probiotics. The active peptides account for 90% of the composition, while the probiotics make up the remaining 10%. The probiotics are modified strains of *S. boulardii*, the modification results in a 25–30% increase in cell wall polysaccharides (the colony-forming unit of the modified probiotics is 6.7 × 10^10^). The peptides are specifically designed and screened with the following sequences of ID13 (Patent No. US11827677): ATCDLLSPFKVGHAACAAHCIARGKRGGWCDGRAVCNCRK.

### 2.3. Experimental Animals and Group Design

A total of 40 3-month-old male Hu sheep (21.65 ± 0.33 kg) were individually weighed, ranked by body weight, and stratified-randomized into two homogeneous groups (*n* = 20 per group): the control group (Con) fed a basal diet, and the experimental group (ATABG) fed the same diet supplemented with 1 g/(sheep·d) ATABG on a dry matter basis. Initial body weight did not differ between groups (21.66 ± 1.37 vs. 21.64 ± 1.54 kg, *p* > 0.05), so weight was not used as a blocking factor.

A 10-day preliminary period was first implemented, during which all Hu sheep were dewormed and acclimated to the experimental environment. Throughout this period, animals in all groups were fed a basal diet without ATABG supplementation, formulated for 30 kg sheep targeting a 200 g/d weight gain according to the Nutritional Requirements of Sheep for Meat in China (NY/T 816-2021) [18], as described in Table 1 of this diet. After the preliminary period, the 63-day formal trial commenced, during which the basal diet was supplemented with ATABG for the ATABG group as described, while the Con group continued receiving the unsupplemented basal diet. Feed was provided three times daily at 8:00, 14:00, and 20:00 throughout both periods. ATABG was mixed with 100 g concentrate using molasses and offered prior to other feedstuffs at each feeding, with 1 g/d evenly distributed across the three meals. Sheep were individually housed in 2.5 m^2^ semi-open pens equipped with free-access salt blocks, free-access water, and natural daylight exposure. Daily environmental monitoring at feeding times recorded temperatures and humidity: 8:00 (19.6 °C, 56% RH), 14:00 (24.6 °C, 52% RH), 20:00 (27.5 °C, 44% RH). Increased ventilation was provided to maintain animal comfort during periods of high ambient temperature.

### 2.4. Sample Collection and Procession

#### 2.4.1. Growth Performance and Apparent Digestibility

During the experimental period, the daily feed amount provided to each sheep group (dry matter content of the provided feed minus that of the leftovers) was recorded to determine the dry matter intake (DMI). Weekly feed refusals from each sheep were combined, processed into approximately 500 g representative refusal samples using the quartering method, and stored at −20 °C. Body weight (BW) was measured on days 1, 21, 42, and 63 of the experiment before the morning feeding, with solid feed withheld for 12 h prior and free access to water maintained throughout the fasting period. For each experimental period (1–21 d, 21–42 d, 42–63 d, and 1–63 d), the average BW was calculated as the mean of the BW at the beginning and end of each period. Daily dry matter intake (DMI, kg/d) was determined by dividing the total DMI of each period by the number of days. DMI as a percentage of BW (% of BW) was calculated as: DMI (% of BW) = DMI (kg/d) ÷ [13× (initial BW + final BW)] × 100%.

As illustrated in Figure 1, the digestive tests commenced on day 21. These tests were performed using the total fecal collection with fecal collection bags, consisting of a 7-day pre-test followed by a 7-day main collection period. In the pre-test, breathable canvas bags were gently secured around the animal’s anus using canvas straps without performing the fecal collection. During the main collection period, the daily feed intake and the amount of leftover feed for each sheep were recorded. All feces from each sheep were collected and weighed daily. Then, 10% of the daily fecal output from each sheep was mixed with 10% sulfuric acid for nitrogen fixation, and the fecal samples were subsequently stored at −20 °C for further evaluation.

#### 2.4.2. Determination Method of Nutrient Content

Before analysis, samples of diet (500 g each), refusal samples (500 g each), and feces (100 g each) were thawed at room temperature and then dried in a forced-air oven at 105 °C for 72 h until a constant weight was achieved. After drying, the samples were ground to pass through a 40-mesh sieve using a high-speed pulverizer to ensure a homogeneous sample for analysis.

The dry matter (DM, method 930.15), crude protein (CP, method 990.03), ether extract (EE, method 920.39), Ca (method 978.02), and P (method 946.06) using the AOAC procedure [19]. Neutral detergent fiber (NDF) and acid detergent fiber (ADF) content were determined following the method of Van Soest et al. [20]. Metabolic energy (ME) was calculated based on the measured nutritional value with the following formula (NY/T 816-2021): ME = 0.046 + 0.820 × (17.211 − 0.135 × NDF).

#### 2.4.3. Rumen Fluid Indicators and Rumen Bacteria Sampling

Sheep were humanely euthanized by electrical stunning followed by jugular vein exsanguination, in accordance with the guidelines for euthanasia of laboratory animals (GB/T 39760-2021) [21]. Immediately after slaughter, two 50 mL portions of ruminal fluid were collected from each sheep by filtering through four layers of cheesecloth. One portion was tested for pH using a pH meter (FE28-standard, Mettler Toledo Co., Ltd., Shanghai, China) with a precision of ±0.01, and the measurement was repeated three times. Enzyme activity in this portion was assayed using a spectrophotometric method with the kit (Suzhou Grace Biotechnology Co., Ltd., Suzhou China), following the manufacturer’s instructions.

The second portion was centrifuged at 4000 rpm for 15 min to collect the supernatant, which was then divided into three 15 mL centrifuge tubes. For NH_3_-N determination, 8 mL of the rumen supernatant was mixed with 2 mL of freshly prepared 25% metaphosphoric acid solution and frozen at −20 °C. For short-chain fatty acids (SCFA) analysis, 4 mL of the rumen supernatant was mixed with 1 mL of the same metaphosphoric acid solution and frozen at −20 °C. SCFAs, including acetate, propionate, and butyrate, were analyzed by HPLC (Agilent 7890A, Santa Clara, CA, USA) with detailed specifications and performance parameters.

An additional 2 mL of ruminal contents were transferred to freezing tubes, immediately frozen in liquid nitrogen, and stored at −80 °C for subsequent DNA extraction and high-throughput sequencing analysis.

Rumen tissue was collected from the dorsal Central Rumen. The tissue was washed three times with 0.9% saline, each time for 5 min, and then immersed in 4% paraformaldehyde for 24 h for fixation. After fixation, the samples were rinsed under running water for 24 h to remove the fixative. Subsequently, paraffin sections were prepared. The paraffin sections were scanned using a Pannoramic SCAN II scanner (Jinan Danjier Electronics Co., Ltd., Jinan China) and analyzed with CaseViewer software (v2.4.0) to measure the length and width of the rumen papillae, as well as the thickness of the rumen muscle.

#### 2.4.4. Extraction of DNA and Sequencing of 16S rDNA

The rumen samples were subjected to total genomic DNA extraction by employing the TGuide S96 Magnetic Soil/Stool DNA Kit (TIANGEN Biotech, Beijing Co., Ltd., Beijing China) following the manufacturer’s instructions. The quality and quantity of the extracted DNA were evaluated through electrophoresis on a 1.8% agarose gel and quantified with a NanoDrop 2000 UV-Vis spectrophotometer (Thermo Scientific, Wilmington, NC, USA). For the amplification of the full-length 16S rRNA gene, primer pairs 27F: AGRGTTTGATYNTGGCTCAG and 1492R: TASGGHTACCTTGTTASGACTT were used. After quantification, amplicons having equimolar concentrations were combined and sequenced on the PacBio Sequel II platform (Beijing Biomarker Technologies Co., Ltd., Beijing, China).

#### 2.4.5. Operational Taxonomic Unit (OTU) Generation Process

The raw reads obtained from sequencing were subjected to filtering and demultiplexing using SMRT Link software (v8.0) with parameters set to minPasses ≥ 5 and minPredictedAccuracy ≥ 0.9, ensuring high-quality circular consensus sequencing (CCS) reads. This parameter setting helps to filter out low-confidence sequences. Subsequently, Lima software (v1.7.0) was used to assign the CCS sequences to corresponding samples based on their barcodes. CCS reads lacking primers and those with lengths outside the range of 1200–1650 bp were discarded after primer identification and quality filtering using Cutadapt (v2.7).

The UCHIME algorithm (v8.1; drive5.com) was employed to detect and eliminate chimera sequences, which are artifacts that can arise during PCR amplification. Clean reads were clustered into operational taxonomic units (OTUs) using USEARCH (v10.0), with sequences showing >97% similarity grouped into the same OTU. OTUs with counts less than 2 in all samples were filtered out to exclude rare and potentially spurious OTUs that might not significantly impact the overall microbial community analysis.

#### 2.4.6. Statistical Analysis

Growth performance, digestibility, and rumen parameters (meeting parametric assumptions) were analyzed using unpaired t-tests, while microbial composition data, including alpha diversity (non-parametric), were tested via Wilcoxon rank-sum tests (SPSS v26.0). Results are reported as mean ± SEM. Growth performance, including DMI, was further evaluated using two-way ANOVA in SPSS (v26.0), considering treatments and time as the two factors, with significance set at *p* < 0.05. Correlation analyses between indicators were conducted using Pearson’s correlation (for variables meeting normality assumptions) and Spearman’s rank correlation (for non-normally distributed or ordinal variables) in Origin (v2022), with significance defined as r > 0.4 and *p* < 0.05.

Taxonomy annotation of the OTUs was conducted using the Naive Bayes classifier in QIIME2 [22] using the SILVA database [23] with a confidence threshold set at 70%. Alpha diversity was assessed to evaluate the complexity of species diversity within each sample using QIIME2 software (v2020). Beta diversity calculations were performed through Non-Metric Multi-Dimensional Scaling (NMDS) to analyze species complexity across samples. Linear discriminant analysis (LDA) combined with effect size (LEfSe) was utilized to identify differentially abundant taxa [24], applying a threshold of *p* < 0.05 and LDA ≥ 2.

## 3. Results

### 3.1. Effect of ATABG on Hu Sheep’s Growth Performance and Apparent Digestibility

Table 2 shows the growth performance of Hu sheep, analyzed with time and groups as two factors. The results indicate that the addition of ATABG did not significantly affect growth performance (*p* > 0.05). The F/G ratio remained unaffected by any factors, including the group, time, or their interaction (*p* > 0.05). Further examination of the data revealed that, during the 1–21-day trial period, a subset of sheep in the Con group exhibited individual body weight loss, leading to a decrease in the mean F/G ratio for this group. However, when considering the entire trial period, the mean F/G ratio of the Con group increased, accompanied by a rise in SEM, indicating increased variability in the data.

As indicated in Table 3, the apparent digestibility of CP, NDF, ADF, and OM was significantly increased with the addition of ATABG, and the intake of NDF and ADF was also significantly increased (*p* < 0.05).

### 3.2. Effect of ATABG on Hu Sheep’s Ruminal Environment and Histomorphology

The rumen fluid and histomorphology indicators are presented in Table 4 and Table 5. The rumen tissue sections are illustrated in Figure 2. The results indicate that the concentration of MCP and cellulase activity were significantly increased with the addition of ATABG (*p* < 0.05). Urease activity in the ATABG group also tended to increase compared to the Con group, with a *p*-value of 0.066. In contrast, the levels of AA and the A/P ratio in the ATABG group tended to decrease compared to the Con group, with *p*-values of 0.097 and 0.098, respectively. Furthermore, rumen pH, RPL, and RMT in the ATABG group tended to be higher than those in the Con group, with *p*-values of 0.096, 0.081, and 0.062, respectively.

The correlation matrix plot of rumen fluid indicators and rumen histomorphology presented in Figure 3 reveals several significant correlations. Among the rumen fluid indicators, NH_3_-N shows a positive correlation with both MCP and LPS. Additionally, MCP demonstrates significant correlations with multiple indicators: it positively correlates with lipase, cellulase, and β-Amylase, while negatively correlating with the A/P ratio. Significant correlations are also observed among lipase, urease, and cellulase, as well as between cellulase and β-Amylase.

In terms of rumen tissue morphology, BA significantly and positively correlates with RPL, RPW, and RMT. Moreover, RPL, RPW, and RMT exhibit positive correlations with one another.

### 3.3. Effect of ATABG on Hu Sheep’s Ruminal Bacterial Relative Abundance

The processing results of the sample sequencing data are presented in Appendix A. The clean circular consensus sequence lengths, sequence length effectiveness, and sequencing read lengths are as follows: For the Con group, the clean circular consensus sequence lengths ranged from 32,884 to 40,496 bp, with sequence length effectiveness between 82.32% and 99.61%, and sequencing read lengths ranging from 1444 to 1467 bp. In contrast, for the ATABG group, the clean circular consensus sequence lengths ranged from 32,667 to 41,375 bp, with sequence length effectiveness between 84.50% and 99.38%, and sequencing read lengths ranging from 1447 to 1473 bp. In Appendix A, sequences were clustered at a 97% similarity threshold, resulting in 4426 operational taxonomic units (OTUs). Among these, 3473 OTUs were shared between the two groups, accounting for 78.47% of the total. The Con group had 302 unique OTUs (6.82%), while the ATABG group had 651 unique OTUs (14.71%).

The average relative abundances of rumen bacteria at the phylum level and genus level are displayed in Figure 4A,B, and Appendix A. The results revealed significant changes in the bacterial composition at the phylum level. Compared with the Con group, the ATABG group showed a significantly higher abundance of Firmicutes, Proteobacteria, Cyanobacteria, and Nanoarchaeota. Specifically, the relative abundance of Firmicutes in the ATABG group was 77.15%, while in the Con group, it was 68.68% (*p* < 0.05). Similarly, for Proteobacteria, the relative abundance in the ATABG group was 2.63% and in the Con group was 1.21% (*p* < 0.05). The Con group, on the other hand, showed a significantly higher abundance of Bacteroidota. Its relative abundance in the Con group was 25.34% compared with 14.38% in the ATABG group (*p* < 0.05). At the genus level, the ATABG group showed a significantly higher abundance of *Ruminococcus*, with a relative abundance of 14.84% in the ATABG group and 7.14% in the Con group (*p* < 0.05).

In Figure 4C, it was observed that the Con group exhibited a significantly higher ACE and Chao1 index compared with the ATABG group (*p* < 0.05), thereby indicating a significantly lower bacterial diversity in the ATABG group. Moving on to the analysis presented in Figure 4D,E, both the PCA and NMDS based on the Bray–Curtis distance matrix analysis revealed no significant difference between the composition of the Con and ATABG groups. The stress for NMDS analysis was 0.0780, indicating a good fit of the NMDS model. Additionally, the 95% confidence ellipses for the two groups overlapped, further supporting the lack of significant compositional differentiation.

In the LEfSe analysis in Figure 5, 11 species were recognized as potential biomarkers from phylum to genus (with LDA scores > 4 and *p* < 0.05). Among the 11 species, the relative abundances of the phylum Bacteroidota, Firmicutes, and the genus *Ruminococcus* are presented in Figure 5B.

### 3.4. Correlation Analysis

Regarding the correlation analysis between the combined group of rumen fluid indicators and rumen histomorphology, and the combined group of apparent digestibility and growth performance in Figure 6A, several significant positive correlations were observed. For instance, pH was significantly negatively correlated with BW, DMI, and ADG, and the correlation coefficients were around −0.47, −0.52, and −0.47, respectively. Additionally, RFI was significantly positively correlated with RPL, with a correlation coefficient of approximately 0.47. Moreover, EED showed significant positive correlations with MCP, lipase, and cellulase. The correlation coefficients for these relationships were 0.44, 0.63, and 0.64, respectively. DMI also exhibited significant positive correlations with RPL, RPW, and RMT, having correlation coefficients of 0.61, 0.67, and 0.58, respectively.

In Figure 6B, a correlation analysis was performed on the top 20 genus-level rumen bacteria in relation to the previously mentioned apparent digestibility, revealing several significant correlations. Notably, multiple genera displayed significant correlations with CPD, with *Ruminococcus* exhibiting a significant positive correlation with CPD (r = 0.50). Additionally, *Ruminococcus* demonstrated a significant positive correlation with A/P, alongside significant negative correlations with the activities of PEP and AMS, with correlation coefficients of 0.48, −0.48, and −0.50, respectively. The genus *Christensenellaceae_R_7_group* showed significant positive correlations with the activities of PEP and AMS, as well as ADG and RFI, with correlation coefficients of 0.44, 0.46, 0.44, and 0.44. Conversely, it exhibited significant negative correlations with pH and CPD (r = −0.45 and −0.51). Furthermore, the genus *Quinella* revealed significant positive correlations with DMI, ADG, and RFI, with correlation coefficients of 0.55, 0.54, and 0.58, while also presenting significant negative correlations with DMD, OMD, and CPD (r = −0.46, −0.47, and −0.66).

## 4. Discussion

Given the concerns regarding antibiotic residues, there has been a shift toward seeking natural, residue-free alternatives to enhance livestock performance [1]. Previous studies involving ruminants have demonstrated the effectiveness of several yeast species, including *Saccharomyces cerevisiae*, *Kluyveromyces marxianus*, *Candida utilis*, and *Saccharomyces boulardii* [17]. However, research on the effects of *S. boulardii* on ruminants has produced mixed results. For instance, Li and Wu [25] found that the addition of *S. boulardii* significantly improved growth performance and feed efficiency in Angus calves. In contrast, Xue et al. [26] reported no difference in the growth performance of Angus calves during a 28-day trial period with the addition of *S. boulardii*. In this study, growth performance parameters, including BW, ADG, and DMI, were primarily affected by time rather than treatment. Previous studies involving *S. cerevisiae* have shown an increase in DMI, attributed to its ability to enhance ruminal conditions that stimulate cellulolysis or affect carbohydrate fermentation [14]. Consistent with findings from DelCurto-Wyffels et al. [27], a negative correlation was observed between rumen fluid pH and BW, DMI, and ADG in this trial. This observation supports the hypothesis that animals may adjust their intake in response to low pH levels [27]. Although the F/G and RFI ratios suggested improved feed efficiency in the ATABG group, no significant difference was found between the two groups. This lack of difference may be attributed to high intra-group variability within the Con group, a factor that could be further validated by increasing the sample size.

Regarding apparent digestibility, the digestive process in the rumen of ruminants is complex, involving the collaborative action of multiple enzymes and microorganisms to degrade different substances [28]. In this study, the ATABG group exhibited significantly higher apparent digestibility of CP, NDF, ADF, and OM compared to the Con group, indicating a positive impact of ATABG on digestion. Furthermore, the MCP concentration in the rumen fluid increased significantly with ATABG supplementation, and there was a trend of increasing urease activity. The improved digestibility and MCP concentration are likely due to more efficient energy and nitrogen release during feed utilization. Nitrogen in rumen fluid exists in various forms, including ammonia, MCP, and undegraded proteins, with MCP and undegraded proteins contributing significantly to the animal’s protein requirements [28]. It is well-established that substrate concentration affects enzyme biosynthesis and activity [29]. In the previous study, researchers fed the bulls with AMPs of cecropin and apidaecin. The results indicated that AMPs could increase the surface area of rumen mucosa and the content of digestive enzymes and VFAs [7]. The observed increase in urease activity and enhanced cellulase activity with ATABG supplementation indicate better utilization of ammonia and carbohydrates. The positive correlations between cellulase, lipase, and urease activities suggest that ATABG facilitates substrate degradation, thereby enhancing enzyme activity and improving MCP synthesis efficiency. In this study, although no differences were observed in β-Amylase activity between the two groups, the positive correlations of β-Amylase activity with MCP concentration and cellulase activity highlight the importance of β-Amylase in MCP synthesis.

Consistent with the conclusion that the addition of *S. boulardii* primarily stabilizes the rumen environment [17], this study observed a trend toward an increase in rumen pH, indicating a nearly significant change. Additionally, AA concentration decreased slightly, and a negative correlation (r = −0.40) was identified between rumen pH and AA concentration. This suggests that the decrease in AA concentration may contribute to the observed increase in pH. In terms of rumen histomorphology, RPL, RPW, and RMT exhibited significant positive correlations with BA. Previous studies have demonstrated that butyrate can promote the growth of ruminal papillae and enhance the expression of genes related to the uptake of VFA in the ruminal epithelium [30]. The results also indicate that rumen papillae respond to variations in the intake of rumen-fermentable organic matter, with significant positive correlations observed between DMI and RPL, RPW, and RMT [31].

The addition of ATABG significantly altered the relative abundances of several phyla and genera within the rumen bacterial composition. At the phylum level, there was a notable increase in Firmicutes, while Bacteroidota exhibited a significant decrease. At the genus level, *Ruminococcus* showed a marked augmentation, as detailed in Appendix A. Firmicutes and Bacteroidota are the predominant phyla in ruminants, playing essential roles in the degradation of cellulose, carbohydrates, and proteins, thereby fulfilling the energy requirements of these animals [32]. Gharechahi and Salekdeh [33] reported that species associated with Bacteroidetes are abundant in genes encoding debranching and oligosaccharide-degrading enzymes, whereas those associated with Firmicutes are rich in cellulases and hemicellulases, all of which are critical for fiber degradation in ruminants. The observed decrease in Bacteroidetes appears to be primarily attributable to the increase in Firmicutes. *Ruminococcus*, a genus within the phylum Firmicutes, demonstrated a significant increase following the addition of ATABG, suggesting that *Ruminococcus* predominantly drove the rise in Firmicutes abundance. LEfSe analysis further underscored the important role of *Ruminococcus*. The previous study of AMPs on bulls significantly increased the abundance of unclassified_f_Ruminococcaceae [7]. In contrast, yeast has several validations in elevating *Ruminalococcus* in the rumen of ruminants [34,35]. In the current research, it is hypothesized that the increase in *Ruminococcus* might be the result of the combined effect of AMPs and *S. boulardii* within the context of ATABG. Previous studies have highlighted the important role of *Ruminococcus* in fiber degradation, protein synthesis, feed efficiency, and overall organism health [13,36,37,38]. The relative abundance of *Ruminococcus* showed a significant positive correlation with CPD (r > 0.40); however, it was not significantly associated with cellulase activity, ADFD, or NDFD in this experiment, suggesting that other factors may be involved. Contrary to the conclusions of Chen et al. [39] regarding the Nanjiang Yellow Goat, the relative abundance of *Ruminococcus* exhibited significant negative correlations with the activities of pepsin and α-amylase. In the study conducted by Yeoman et al. [38], *Ruminococcus* employs multiple mechanisms to degrade plant cell walls, including cellulosomal carbohydrate-active enzymes, type IV pili, and CBM37-harboring components. Given that α-amylase is classified as one of the cellulosomal carbohydrate-active enzymes, the observed negative correlation between the relative abundance of *Ruminococcus* and α-amylase activity may stem from substrate competition. Furthermore, in addition to shifts in the predominant phyla, Proteobacteria, Cyanobacteria, and Nanoarchaeota demonstrated significant increases following the addition of ATABG. These phyla are adaptable and can flourish in diverse environments, fulfilling various ecological roles. Previous studies have indicated that the phylum Proteobacteria correlates positively with dissolved organic carbon and plays a crucial role in bioflocs [40]. Additionally, Proteobacteria have been positively correlated with fiber intake [41]. Cyanobacteria are recognized for their critical functions, most notably their ability to perform photosynthesis, during which they utilize light energy to convert carbon dioxide and water into organic compounds, releasing oxygen as a byproduct [41]. Another significant function of Cyanobacteria is nitrogen fixation, allowing them to convert atmospheric nitrogen into a form usable by other organisms [41]. The increase in these phyla may have enhanced nitrogen and carbon utilization.

Several correlations between genera and epigenetic indicators were observed. *Rikenellaceae_RC9_gut_group* showed significant positive correlations with meat fat and volatile fatty acids in Tan sheep [42]. In the study by Jiao et al. [43], the abundance of *Rikenellaceae_RC9_gut_group* decreased as the pH increased from 6.0 to 6.6, and in this study, a similar negative correlation between pH and *Rikenellaceae_RC9_gut_group* was observed. However, the previous study noted that the *Rikenellaceae_RC9_gut_group* can enhance lipid metabolism and facilitate carbohydrate and protein fermentation [44]. In contrast, the current study yielded the opposite result. After further investigation, we found that the reason might be that adding ATABG increased CPD and the abundance of *Ruminococcus*, leading to negative correlations between CPD and other genera. The *Christensenellaceae_R_7_group*, which shares similar functions with the Christensenellaceae family, is an effective sugar-fermenting group capable of converting glucose into acetic and butyric acids while degrading cellulose [45]. Unlike the study by Gu et al. [46], which reported positive correlations between the relative abundance of *Christensenellaceae_R_7_group* and both CPD and ADFD in Min Dong male goats, our study found a significant negative correlation between CPD and *Christensenellaceae_R_7_group*, reinforcing our earlier hypothesis. The genus *Quinella*, categorized as an iconic rumen bacterium, was first documented in 1913. However, as noted by Kumar et al. [47], there is limited and conflicting information regarding its physiological characteristics. Genome analysis of *Quinella* indicates that it lacks the ability to independently degrade polysaccharides, instead relying on other rumen microorganisms for this process and using the resulting products for its growth [47]. A previous study by [48] also emphasized the significant role of Quinella, reporting negative correlations between its relative abundance and both altitude and rumen fermentation parameters. In our study, *Quinella* exhibited significant positive correlations with DMI, ADG, and RFI, suggesting its potential role in assessing feed efficiency.

## 5. Conclusions

Supplementation with the compound microecological preparation ATABG improved nutrient digestibility and rumen function in Hu sheep by enhancing key digestive parameters and modulating the ruminal microbiota. Apparent digestibility of crude protein, NDF, ADF, and organic matter was significantly elevated, reflecting improved feed utilization. Rumen fluid analysis showed increased microbial protein concentration and cellulase activity, indicative of enhanced nitrogen metabolism and fiber degradation. Microbiota profiling revealed a shift toward a Firmicutes-dominant community with enriched *Ruminococcus*—a key genus for fiber degradation—and reduced Bacteroidota. These changes were associated with improved cellulase activity and positive correlations between *Ruminococcus* abundance and crude protein digestibility. Collectively, ATABG optimizes rumen fermentation by promoting functional bacteria and nitrogen utilization, offering a promising approach to enhance nutrient efficiency in ruminant production.

## Figures and Tables

**Figure 1 microorganisms-13-00999-f001:**
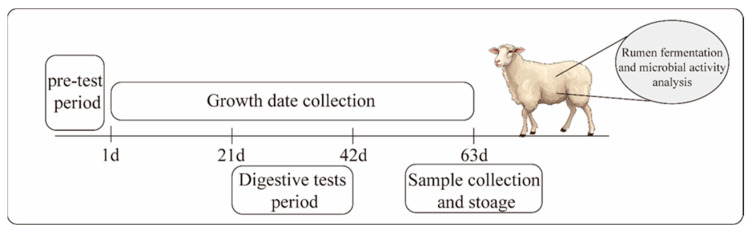
Schematic diagram of the experimental process.

**Figure 2 microorganisms-13-00999-f002:**
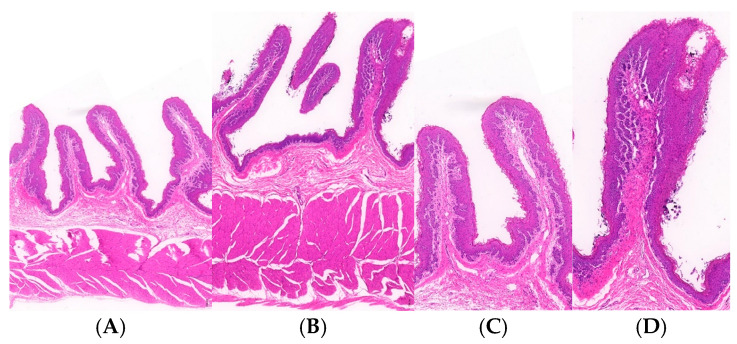
Rumen tissue morphology of two groups. Rumen tissue morphology of the Con group at (**A**) 4.0× and (**C**) 10.0×; Rumen tissue morphology of the ATABG group at (**B**) 4.0× and (**D**) 10.0×.

**Figure 3 microorganisms-13-00999-f003:**
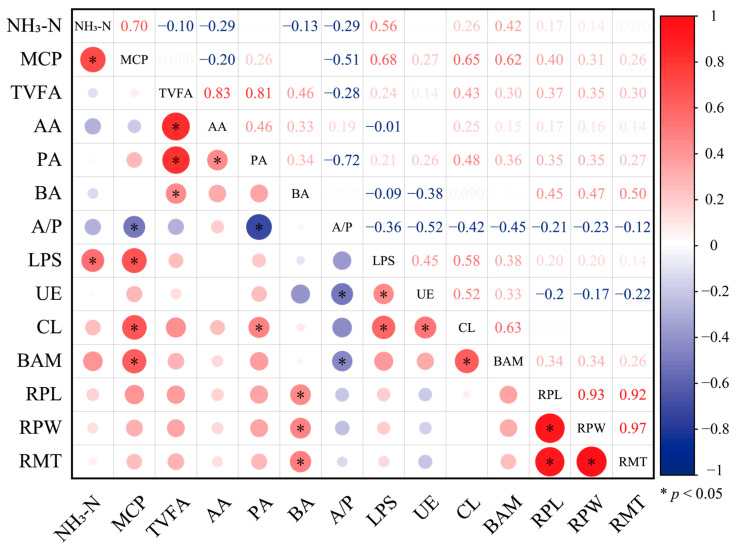
Correlation matrix plot of rumen fluid indicators and rumen histomorphology. The Pearson correlation between two indicators is represented by a graph or a number at the intersection of the two indicators. The size of the circle in the lower left corner indicates the magnitude of the correlation between the indicators. Warm colors represent positive correlations, while cold colors represent negative correlations. Additionally, an asterisk (*) is used to denote a significant correlation (*p* < 0.05). In the upper right corner of the figure, corresponding to each graphic, the Pearson’s correlation coefficient is presented to indicate the correlation value between the two indicators. TVFA, total volatile fatty acids; AA, acetic acid; PA, propanoic acid; BA, butyric acid; A/P, acetic acid to propionic acid ratio; MCP, microbial protein; LPS, lipase; UE, urease; CL, cellulase; BAM, β-Amylase; RPL, rumen papillae length; RPW, rumen papillae width; RMT, rumen muscular layer thickness.

**Figure 4 microorganisms-13-00999-f004:**
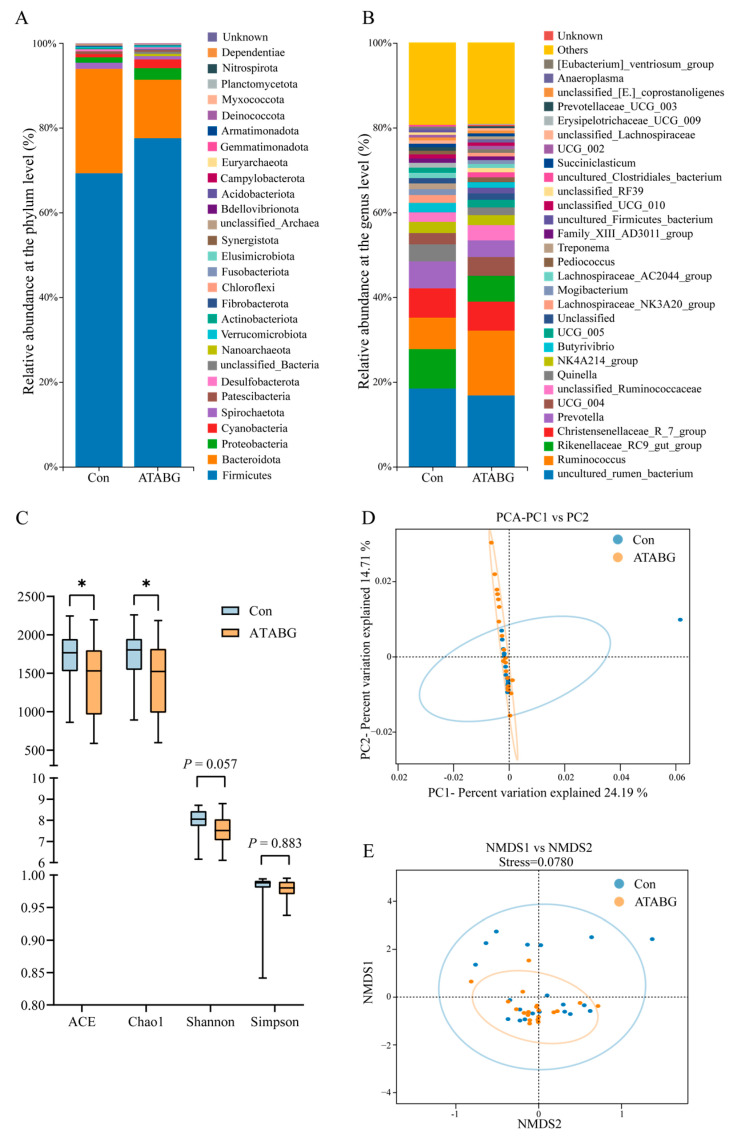
Average relative abundance of rumen microbial taxa and community diversity. (**A**) and (**B**) display the columnar composition plots of the average relative abundance of rumen bacteria at the phylum and genus levels. (**C**) presents the analysis of alpha diversity in the microbial composition of the two groups, with the asterisk (*) denoting a significant difference (Wilcoxon rank-sum tests, *n* = 20, *p* < 0.05). (**D**) shows the PCA analysis and (**E**) the NMDS analysis of the ruminal microbial composition of sheep based on the Bray–Curtis distance matrix analysis in the two groups.

**Figure 5 microorganisms-13-00999-f005:**
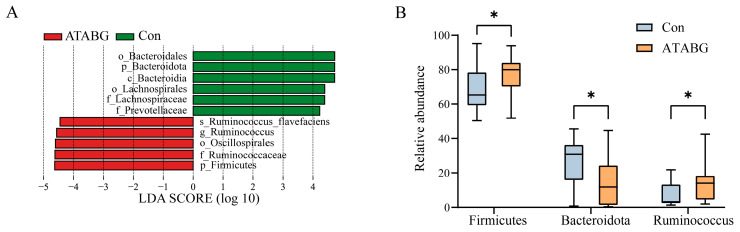
LEfSe analysis of rumen bacterial composition. (**A**) displays LDA analysis, the vertical coordinates as the categorical units with significant differences between groups, and the horizontal coordinates display the logarithmic score values of the LDA analysis for each categorical unit. Longer lengths on the horizontal axis indicate more significant differences. The threshold for LDA is set to 4. The distribution of the relative abundance of differential categories across different phylum levels and genus level is shown (**B**), where solid and dashed lines represent the mean and median relative abundance of each categorical unit in each subgroup. The asterisk (*) denotes a significant difference (Wilcoxon rank-sum tests, *n* = 20, *p* < 0.05).

**Figure 6 microorganisms-13-00999-f006:**
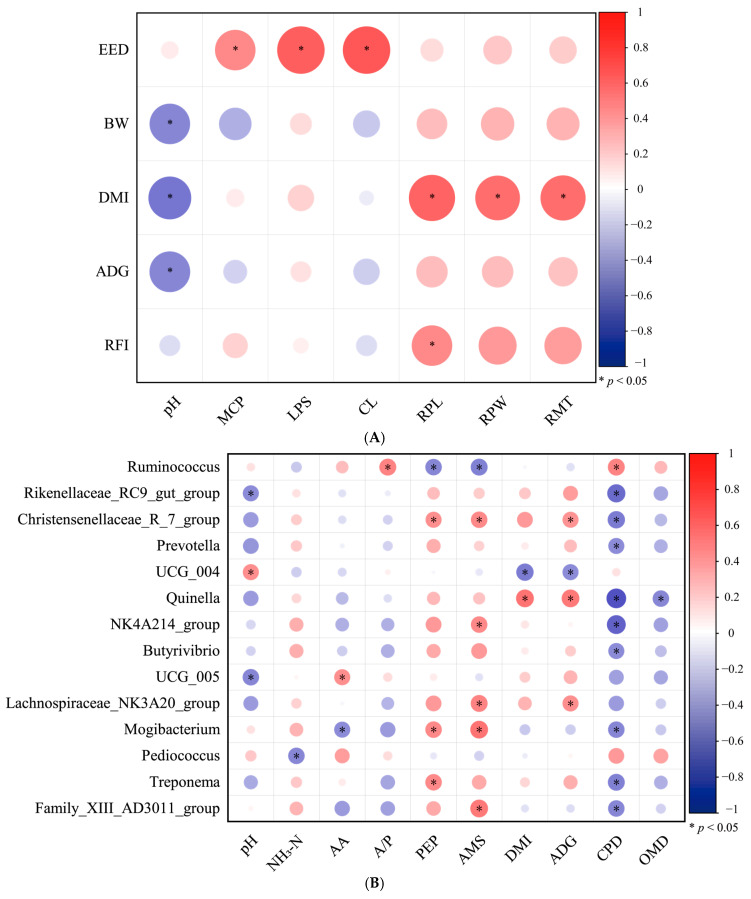
Correlation matrix plot presenting the associations between the indicators. (**A**) displays the Pearson correlation between the combined group of rumen fluid indicators and rumen histomorphology, and the combined group of growth performance and the apparent digestibility of nutrients. (**B**) displays the Spearman correlation between the combined group of rumen fluid indicators and rumen histomorphology, and the combined group of growth performance and the apparent digestibility of nutrients. Warm colors represent positive correlations, while cold colors represent negative correlations. Additionally, an asterisk (*) is used to denote a significant correlation (*p* < 0.05). EED, ether extract digestibility; BW, Body weight; ADG, average daily gain; DMI, dry matter intake; RFI, residual feed intake; MCP, microbial protein; LPS, lipase; CL, cellulase; RPL, rumen papillae length; RPW, rumen papillae width; RMT, rumen muscular layer thickness.

**Table 1 microorganisms-13-00999-t001:** Diet composition and nutritional level (dry matter basis).

Ingredients	Content, %	Nutrient Levels ^2^	Content ^3^, %
Alfalfa	35.00	ME, MJ/kg	9.22
Cornstalk	25.00	CP	13.28
Corn	28.00	NDF	44.59
Soybean meal	5.00	ADF	23.60
Wheat bran	4.00	EE	13.95
molasses	1.00	NFC	22.29
NaCl	1.00	NFC/NDF	49.99
Premix ^1^	1.00	Ca, g/kg	0.84
Total	100.00	TP, g/kg	0.24

^1^ Premix provided the following per kilogram of the diet: VA 1400 IU, VD_3_ 500 IU, VE 50 mg. Fe 10 mg, Cu 12.5 mg, Mn 135 mg, Zn 100 mg, Co 0.5 mg, I 1.5 mg, Se 0.3 mg; ^2^ Nutrients were determined on a dry matter basis at 105 °C; ME, metabolic energy; CP, crude protein; NDF, neutral detergent fiber; ADF, acid detergent fiber; EE, ether extract; NFC, non-fibrous carbohydrate; TP, total phosphorus; ^3^ The metabolic energy and non-fibrous carbohydrates in the nutritional level were calculated. The calculation was done in reference to the Nutritional Requirements of Sheep for Meat in China (NY/T 816-2021). The formula for calculating metabolic energy is ME = 0.046 + 0.820 × (17.211 − 0.135 × NDF). And the formula for calculating non-fibrous carbohydrates is NFC = 1 − (CP % + EE % + Ash % + NDF %). The rest were measured values.

**Table 2 microorganisms-13-00999-t002:** Effects of ATABG supplementation on the growth performance of fattening Hu sheep.

Items	Time	Groups		*p*-Value		
		Con	ATABG	Time	Groups	T×G
BW, kg	1 d	21.66 ± 1.37	21.64 ± 1.54	<0.01	0.650	0.924
	21 d	25.32 ± 1.66	25.54 ± 1.71			
	42 d	29.17 ± 1.79	28.70 ± 1.77			
	63 d	33.64 ± 1.96	33.07 ± 1.80			
ADG, g	1–21 d	174.29 ± 10.10	185.48 ± 8.70	0.024	0.384	0.490
	21–42 d	183.43 ± 7.86	150.64 ± 6.91			
	42–63 d	212.76 ± 8.32	207.93 ± 6.23			
	1–63 d	190.16 ± 7.35	181.35 ± 5.84			
DMI, g/d	1–21 d	1070.83 ± 45.50	1035.59 ± 32.03	<0.01	0.140	0.968
	21–42 d	1293.02 ± 44.87	1216.76 ± 31.25			
	42–63 d	1645.78 ± 130.57	1541.67 ± 104.92			
	1–63 d	1336.54 ± 41.80	1264.67 ± 39.84			
DMI, % of BW	1–21 d	4.57 ± 0.19	4.42 ± 0.14	0.050	0.192	0.990
	21–42 d	4.77 ± 0.17	4.54 ± 0.16			
	42–63 d	5.28 ± 0.44	4.98 ± 0.32			
	1–63 d	4.84 ± 0.12	4.64 ± 0.15			
F/G	1–21 d	5.80 ± 2.05	6.60 ± 1.84	0.156	0.761	0.340
	21–42 d	7.89 ± 1.72	9.50 ± 2.43			
	42–63 d	8.78 ± 2.26	7.54 ± 1.55			
	1–63 d	9.26 ± 3.26	7.08 ± 0.93			

BW, body weight; ADG, average daily gain; DMI, dry matter intake; F/G, feed-to-gain ratio; T×G, interaction of time and groups. Values are presented as mean ± SEM (*n* = 20). Differences between the two groups were evaluated using two-way ANOVA, with statistical significance set at *p* < 0.05.

**Table 3 microorganisms-13-00999-t003:** Effects of ATABG supplementation on apparent digestibility of fattening Hu sheep.

Items		Groups		*p*-Value
		Con	ATABG	
DM	Intake, g	876.47 ± 16.08	886.13 ± 13.54	0.649
	Digestibility, %	56.86 ± 1.09	58.45 ± 3.28	0.650
OM	Intake, g	510.74 ± 14.16	556.12 ± 13.22	0.076
	Digestibility, %	63.05 ± 0.98	66.65 ± 1.16	0.010
CP	Intake, g	65.37 ± 2.07	69.90 ± 2.16	0.139
	Digestibility, %	56.07 ± 1.21	59.33 ± 1.46	0.034
EE	Intake, g	95.78 ± 2.58	98.60 ± 2.10	0.401
	Digestibility, %	78.25 ± 1.34	79.73 ± 1.07	0.252
NDF	Intake, g	203.68 ± 7.04	226.28 ± 7.49	0.034
	Digestibility, %	51.99 ± 1.14	57.25 ± 1.68	0.011
ADF	Intake, g	106.16 ± 4.36	122.15 ± 4.39	0.014
	Digestibility, %	51.11 ± 1.38	58.51 ± 2.20	<0.01

DM, dry matter; OM, organic matter; CP, crude protein; EE, ether extract; NDF, neutral detergent fiber; ADF, acid detergent fiber. Values are presented as mean ± SEM (*n* = 20). Differences between the two groups were evaluated using an independent-samples *t*-test, with statistical significance set at *p* < 0.05.

**Table 4 microorganisms-13-00999-t004:** Effects of ATABG supplementation on the ruminal environment of fattening Hu sheep.

Items	Groups		*p*-Value
	Con	ATABG	
pH	6.67 ± 0.03	6.73 ± 0.03	0.096
TVFA, mmol/L	71.05 ± 2.59	68.91 ± 2.26	0.537
acetic acid, mmol/L	40.47 ± 1.51	37.18 ± 1.21	0.097
acetic acid, % of TVFA	57.15 ± 1.17	54.16 ± 0.93	0.052
propanoic acid, mmol/L	19.56 ± 1.24	20.46 ± 1.28	0.615
propanoic acid, % of TVFA	27.25 ± 1.07	29.26 ± 1.08	0.193
butyric acid, mmol/L	11.02 ± 0.57	11.26 ± 0.23	0.695
butyric acid, % of TVFA	15.56 ± 0.84	16.58 ± 0.50	0.322
A/P	2.18 ± 0.12	1.92 ± 0.10	0.098
NH_3_-N, mg/dL	16.58 ± 0.88	15.66 ± 0.57	0.384
MCP, mg/dL	9.34 ± 0.47	10.81 ± 0.39	0.020

MCP, microbial protein; TVFA, total volatile fatty acids; A/P, acetic acid to propionic acid ratio. Values are presented as mean ± SEM (*n* = 20). Differences between the two groups were evaluated using an independent-samples *t*-test, with statistical significance set at *p* < 0.05.

**Table 5 microorganisms-13-00999-t005:** Effects of ATABG supplementation on the ruminal enzyme activity and rumen histomorphology of fattening Hu sheep.

Items	Groups		*p*-Value
	Con	ATABG	
lipase, mol/min/mL	2.24 ± 0.21	2.26 ± 0.22	0.926
urease, ug/min/mL	0.51 ± 0.10	0.80 ± 0.12	0.066
cellulase, ug/min/mL	4.18 ± 0.57	6.56 ± 0.97	0.041
pepsin, U/mL	79.26 ± 14.28	68.73 ± 9.50	0.543
α-amylase, ug/min/mL	37.75 ± 4.27	27.81 ± 4.12	0.102
β-amylase, ug/min/mL	54.15 ± 5.59	64.46 ± 4.76	0.168
RPL, mm	1.65 ± 0.09	2.00 ± 0.03	0.081
RPW, mm	0.41 ± 0.03	0.39 ± 0.03	0.625
RMT, mm	1.36 ± 0.05	1.49 ± 0.04	0.062

RPL, rumen papillae length; RPW, rumen papillae width; RMT, rumen muscular layer thickness. Values are presented as mean ± SEM (*n* = 20). Differences between the two groups were evaluated using an independent-samples *t*-test, with statistical significance set at *p* < 0.05.

## Data Availability

Data are contained within the article and Appendix A. The raw datasets generated during the current study are available in the NCBI (https://www.ncbi.nlm.nih.gov/bioproject/PRJNA1179656, accessed on 12 March 2025) repository.

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
