# Peer review of "Effects of Compound Microecological Preparation Supplementation on Production Performance and Nutrient Apparent Digestibility in Hu Sheep from the Rumen Perspective"

_microorganisms, 2025, doi:10.3390/microorganisms13050999_

Round 1

Reviewer 1 Report

Comments and Suggestions for Authors The paper is original and interesting to a wide range of readers. The methodology is correct. The analyses are adequate and there are enough of them. The conclusions are presented briefly and clearly and confirm the research hypothesis. There are enough references and it is cited correctly. I have no suggestions for improvement and I think the paper can be published in this form.

Author Response

Thank you very much for your comments, which we have revised based on the comments of the other reviewers.

Reviewer 2 Report

Comments and Suggestions for Authors

The manuscript presents relevant data on the effects of dietary treatments. But methodological descriptions lack detail, particularly regarding diet formulation, nutrient requirements, dosing, sampling procedures, and experimental design. The absence of key tables (e.g., Table 1) and the unclear reporting of some variables (e.g., DMI, tissue sampling, nutrient intake) compromise the interpretation of the results. Statistical analysis choices (parametric vs. non-parametric) need to be justified, and the construction of a predictive model for dry matter intake is questioned given its limited relevance and performance. Tables and figures should be revised for clarity and consistency, with proper use of abbreviations and footnotes. The conclusions should be improved to better reflect the study’s findings rather than summarize results. Overall, the manuscript has potential but requires substantial revision to enhance scientific rigor and readability.

L21: My recommendation is to use abbreviations in major cases as well. Basic diet or basal diet?

L23-36 Please rewrite, be more concise with the results, and finish with a short conclusion

L37-38: It is not common to use the same words used in the title

L86: Please don't use flora, check the correct Vocabulary https://microbiomejournal.biomedcentral.com/articles/10.1186/s40168-015-0094-5

L90-92: This is not part of an introduction, can be included in the abstract as a conclusion, or deleted

L106: Please check the term basic diet. In this section is not clear how the ATABG was included in the diet. Is this blend stable on the environment? Does it need to be prepared every day?

L111: What nutritional requirements were considered? How was this dosage determined?

L112 There is no Table 1 in the manuscript. Thus, it is not possible to know the diet composition or the ingredients. What meaning (NY/T 816-2021)?

L120: According to sampling details, this should be the 2.4.2. section

L128: This should be the 2.4.1 section

L129-130: I think that the amount provided is not enough to obtain DMI. I mean, offered is not equal to ingested.

L130-131: Was it after a solid and/or liquid fasting period?

L133-134: Please explain what meaning of Bo, B1, B2, and B3. How do you use the DMI data, I mean, daily, or as a means by period, etc.

L138: How did you use a bag adapted or a metabolic cage?

L140: This is normal in urine samples to avoid N volatilization, but I have never read about feces. The authors, please explain to me why they used this procedure in their trial.

L158: Please add information about tissue collection.

L191: Rumen fermentation parameters?

L193: Do all data meet the assumptions of parametric tests?

L194-195: What about DMI data? Why wasn't the time effect also considered?

L196-197: I don't understand why authors used a non-parametric correlation test, while all mean-comparison methods were parametric.

L210-211: How? I also would like to see the DMI expressed as BW of sheep (% of BW)

L217: The structure of table 2 needs to be modified, is too difficult to read. A simple table containing all variable answers in the lines and using the columns to report the mean of each treatment, SEM, and 3 columns containing each p-value could show clearly the results.

L216-217: Why do authors construct a prediction model for dry material intake? This is not the main focus of the study. In addition, the model is too poor, my recommendation is to remove this model and focus on your trial results!

L230-231: Why did you not report nutrient intake?

L237-238: I am very surprised, because in M&M there is nothing about these measurements

L244: Considering that the total VFA did not have differences, the authors also need to express individual VFA as a proportion of the total VFA.

L246: Some abbreviations as RPL, PRE, and RMT, are not described in the footnote table. Each table and figure needs to be auto-explanatory.

L306: Does it show the mean or the median? 

L309: by xxx test. What does the stress report?

L321-322: Please indicate to the reader what meaning the *

L348-350: Due to the few significant correlations found, it may be easily presented in Figure 4. Can remove variables that did not show any significant correlation. 

L491-499: Conclusions seem like a short abstract of your results. It can be improved.

Author Response

Thank you very much for your comments. We have responded to your comments point by point, and have combined with other reviewers' comments to make changes, which are marked in red. The specific responses are in the annexe.

Reviewer 3 Report

Comments and Suggestions for Authors

This article provides information on the effects of compound microecological preparations supplementation on production performance, nutrient apparent digestibility and ruminal parameters in Hu Sheep. It is in general appropriately organized, carried out and written, however there are some points that should be corrected or clarified.

Please check comments and corrections in the attached file.

Comments on the Quality of English Language

The English could be improved to more clearly express the research

Author Response

(The authors gave the same response as above.)

Reviewer 4 Report

Comments and Suggestions for Authors

Line 20 - insert the average weight of the animals.

Line 22 - What is the ratio of bulk: concentrate in the diets? What was the experimental period? Present a brief sentence about what was collected after slaughter.

Line 37 - 38 - Remove the words contained in the title from the keywords

Lines 90 - 92 - Unnecessary. The introduction should end only with the objective of the research. This information is an expected result and should not be included here.

Line 93 - The location of the experiment should be described. Information about the climate, temperature, humidity, and precipitation is welcome, due to the scope of readers that the research can reach and the climate diversity. Factors that can bioclimatically affect the animal's feeding behavior.

Lines 108-109 - What was the facility like where the animals were confined? Was it in an open or closed shed? Or was it in full sun? What were the dimensions of the stalls? Did they have fixed feeders and drinkers? How were the animals distributed in each treatment? Was there a standardization of weights so that the treatments became homogeneous? If the animals were weighed and then distributed in each treatment, the weight of the animals is a factor for blocking.

Line 112 - What ingredients were included in the diets? What is the ratio of bulk: concentrate?

Table 1 with the diets offered and the nutritional composition of the diets should be included. Add the metabolizable energy of the diets. It is also important that the composition of the ingredients of the formulated diets be included.

Line 113 - What happened during the adaptation period? Were the animals vaccinated and dewormed? What management was carried out?

Line 115 - How many liters of water were offered? Was water consumption monitored? How did the authors quantify the evaporated water?

Lines 116 - 117 - How were the environmental conditions monitored? Were the animals under heat stress due to the environmental conditions?

Lines 121 - 127 - I believe this item should be presented after 2.4.2, due to the details presented in this item. All methods should be better detailed. How long were the samples collected? How many grams of sample were used for these analyses? How were the samples processed for the analyses? The total carbohydrate and non-fibrous carbohydrate contents should be presented.

Line 145 - When was the slaughtering carried out? How was it carried out? What criteria were followed? How was the ruminal fluid collected? Details are important

Line 151 - Change VFA to short-chain fatty acids (SCFA)

Line 158 - Was ruminal tissue collected from all rumen sacs? Identify where the collections were performed.

Describe in the tables which mean test was applied in relation to the significance level

I suggest that the authors enlarge the images for better visualization

Author Response

(The authors gave the same response as above.)

Round 2

Reviewer 2 Report

Comments and Suggestions for Authors

Authors have improved the manuscript according to my comments.